# Comparison of Ruptured Intracranial Aneurysms Identification Using Different Machine Learning Algorithms and Radiomics

**DOI:** 10.3390/diagnostics13162627

**Published:** 2023-08-09

**Authors:** Beisheng Yang, Wenjie Li, Xiaojia Wu, Weijia Zhong, Jing Wang, Yu Zhou, Tianxing Huang, Lu Zhou, Zhiming Zhou

**Affiliations:** 1Department of Radiology, The Second Affiliated Hospital of Chongqing Medical University, Chongqing 400000, China; yangbs@cqmu.edu.cn (B.Y.); lwj120932@163.com (W.L.); 304831@hospital.cqmu.edu.cn (X.W.); zhongwj@cqmu.edu.cn (W.Z.); wangjinghan951127@163.com (J.W.); zy279952@outlook.com (Y.Z.); 13981974832@163.com (T.H.); 17725127237@163.com (L.Z.); 2Department of Radiology, The Third Affiliated Hospital of Chongqing Medical University, Chongqing 400000, China

**Keywords:** intracranial aneurysm, rupture, radiomics, machine learning, computed tomography angiography

## Abstract

Different machine learning algorithms have different characteristics and applicability. This study aims to predict ruptured intracranial aneurysms by radiomics models based on different machine learning algorithms and evaluate their differences in the same data condition. A total of 576 patients with intracranial aneurysms (192 ruptured and 384 unruptured intracranial aneurysms) from two institutions are included and randomly divided into training and validation cohorts in a ratio of 7:3. Of the 107 radiomics features extracted from computed tomography angiography images, seven features stood out. Then, radiomics features and 12 common machine learning algorithms, including the decision-making tree, support vector machine, logistic regression, Gaussian Naive Bayes, k-nearest neighbor, random forest, extreme gradient boosting, bagging classifier, AdaBoost, gradient boosting, light gradient boosting machine, and CatBoost were applied to construct models for predicting ruptured intracranial aneurysms, and the predictive performance of all models was compared. In the validation cohort, the area under curve (AUC) values of models based on AdaBoost, gradient boosting, and CatBoost for predicting ruptured intracranial aneurysms were 0.889, 0.883, and 0.864, respectively, with no significant differences among them. Of note, the performance of these models was significantly superior to that of the other nine models. The AUC of the AdaBoost model in the cross-validation was within the range of 0.842 to 0.918. Radiomics models based on the machine learning algorithms can be used to predict ruptured intracranial aneurysms, and the prediction efficacy differs among machine learning algorithms. The boosting algorithms might be superior in the application of radiomics combined with the machine learning algorithm to predict aneurysm ruptures.

## 1. Introduction

Intracranial aneurysms, which are a life-threatening “untimed bomb” inside the skull potentially causing an aneurysmal subarachnoid hemorrhage (aSAH), affect approximately 1–3% of the adult population [1,2,3]. However, a large proportion of detected but unruptured aneurysms remain asymptomatic and unruptured during a lifelong follow up [4]. The preventive treatment for such aneurysms is risky and expensive, though once ruptured, the outcome is catastrophic [2]. The accurate and timely assessment of the rupture risk of intracranial aneurysms would be of great significance to clinical practice.

In clinical practice, scales based on the clinical data and morphology of the aneurysm, such as PHASES or ELAPSS scores [5,6], play an important role in assessing the risk of rupture. With the improvement in efficacy encountering a bottleneck, the topic of interest introduced new features, such as radiomics [7]. Radiomics has the advantage of using features that cannot be obtained by regular observations and can achieve a more comprehensive and detailed quantification of features. Recent studies have shown that radiomics features were different between ruptured and unruptured intracranial aneurysms [8,9,10]. Furthermore, most studies have shown that the use of radiomics features may improve the rupture prediction performance of intracranial aneurysms [8,9]. However, the future research must to explore how to analyze radiomics features to better capture aneurysm characteristic information to more effectively predict ruptured aneurysms.

Machine learning can not only process complex data, including morphologic or radiomics features or fusion features from different types of features medical imaging methods [11,12,13], but can also identify trends and patterns that humans may miss. In addition, the traditional machine learning method has the advantages of reduced training time and is more suitable for a small data size than deep learning. The model and results based on traditional machine learning methods are easy to understand and interpret, and the application is relatively flexible, such as algorithm hybridization [14]. With the combination of radiomics and machine learning, the prediction of intracranial aneurysm ruptures has achieved accumulating salient results [15,16,17,18], which achieved the best AUC of 0.86 [18]. However, the hidden possibilities in diverse machine learning algorithms have been ignored. In a review of previous work [15,16,17,18] and a brief overview of the application of machine learning, we noticed that there were many machine learning algorithms suitable for the prediction of aneurysm ruptures, which reminded us that by using the dataset, including the population of patients and radiomics features derived from them, a proper machine learning algorithm for further analysis could be developed.

Thus, in this study, after extracting the radiomics features of intracranial aneurysms from computed tomography angiography (CTA) images, our main interests focus on the efficacy of different algorithms combined with radiomics for predicting aneurysm ruptures with the same baseline parameters. We aim to predict ruptured intracranial aneurysms using different machine learning algorithms combined with radiomics and evaluate their differences.

## 2. Materials and Methods

The design of this study is shown in Figure 1.


**Participants**


This experiment was approved by the ethics standards committee on human experimentation of the Second Affiliated Hospital of Chongqing Medical University, and written informed consent was obtained from each participant.

Patients diagnosed with intracranial aneurysms by CTA or digital subtraction angiography (DSA) in the two centers of our hospital between 2015–2021 were included in this study. The exclusion criteria included: (1) secondary intracranial aneurysms of primary vascular disease or intracranial aneurysms combined with intracranial vascular diseases (such as Moyamoya disease, arteriovenous malformations, and autoimmune-related vascular disease, etc.); (2) multiple or fusiform intracranial aneurysms; (3) intracranial aneurysms with a maximum diameter < 2 mm on CTA images; (4) intracranial aneurysms that could not be differentiated from infundibulum on CTA images; (5) poor quality of CTA images (motion artifacts, delayed scanning, etc.); and (6) surgical or interventional therapy of intracranial aneurysms before the CTA examination. Finally, a total of 739 patients were included in this study; 192 who suffered from SAH during the follow up were diagnosed as having ruptured intracranial aneurysms. The demographic data of all patients including a history of hypertension and SAH was recorded.


**Image Acquisition and Analysis**


CTA examinations were performed on multislice spiral CT scanners (Aquilion ONE, Canon medical Systems, Japan; Somatom Definition Force, Siemens Healthcare, Germany), with the following scanning protocol: a tube voltage of 110–120 kV, 200–250 mA, a layer thickness of 1.0 mm, a layer spacing of 0.7 mm, and a matrix of 512 × 512. The contrast agent was iohexol solution, the total dose was 150–300 mg/kg of iodine, and the injection flow rate was 4.5–5.0 mL/s.

Given the images derived from multiple CT scanners with different parameters, the images were preprocessed as follows: (1) the CTA images were resampled with the voxel size 1.0 × 1.0 × 1.0 mm^3^; (2) gray-level discretization with the original intensities were resampled with a fixed bin width (256 bins); and (3) the characteristics of intracranial aneurysms were viewed in a fixed window (level, 50 Hounsfield unit [Hu]; width, 110 Hu) on CTA images. Image analysis of intracranial aneurysms was performed independently by three radiologists who were blinded to the clinical status of the patients to provide a consensus as to the final interpretation. Multiplanar reformation technology was applied when necessary to measure the long diameter and for determining the location of the intracranial aneurysm. In addition, the PHASES scores were analyzed to evaluate the general risk of rupture of intracranial aneurysms [5].


**Simple Random Sampling**


Considering the influence of sample imbalance between the ruptured and unruptured intracranial aneurysm groups, we randomly selected 384 patients from the sample population with unruptured intracranial aneurysms using the random seed of 68,439. The ratio of individuals in the unruptured intracranial aneurysm group to ruptured intracranial aneurysm group was 2:1. In order to test whether the distribution of the random sample was consistent with that of the population, Poisson, negative binomial, normal, gamma, and generalized Pareto distribution tests were conducted on the PHASES scores of the random sample and population [5,6]. Then, we found that the PHASES scores of the population presented the minimum standard errors and were in agreement with the negative binomial and Poisson distributions, with scores of 0.047 and 0.049, respectively, and those of the corresponding test sample (384 patients) were 0.060 and 0.070, respectively. In addition, we compared these models with the other distribution models; the corresponding sample remained at the minimum value. Thus, it can be considered that the 384 randomly selected patients could represent the sample population with unruptured intracranial aneurysms, and the corresponding sample was chosen as the unruptured intracranial aneurysm group for the analysis. 

Finally, 576 patients with intracranial aneurysms were retrospectively reviewed and they were randomly divided into training (n = 403) and validation (n = 173) cohorts in a ratio of 7:3 by computer software-generated random numbers.


**Radiomics Analysis and Models**


Firstly, the volume of interests (VOIs) of the intracranial aneurysm group was manually sketched slice by slice on CTA images by a trained radiologist using ITK-SNAP software (version 3.8.0) and double-checked by another radiologist. Then, a total of 107 radiomics features were extracted automatically from each VOI using PyRadiomics software (version 3.0.1). Then, inter-observer and intra-observer reproducibility analyses were performed to assess the stability of radiomics features (see Appendix A).

Harmonization in the feature domain was performed according to the previous research before further feature selection [19,20], which is described in the Appendix A. Then, an independent sample test and elastic network regression analysis were performed to choose the optimal radiomics features related to ruptured intracranial aneurysms. Finally, the optimal radiomics features and 12 common machine learning algorithms (including the support machine learning (SVM), decision-making tree, eXtreme gradient boosting (XGB), Gaussian Naive Bayes (GNB), logistic regression, random forest, k-nearest neighbor (KNN), bagging classifier, AdaBoost, gradient boosting, light gradient boosting machine (LGBM), and CatBoost) were used to construct models for predicting intracranial aneurysm ruptures in the training cohort, and then validated in the validation cohort. The calibration curves were plotted to assess the calibration ability of the 12 machine learning models. The area under curve (AUC), sensitivity, specificity, positive predictive value, negative predictive value, and accuracy of each model in the validation cohort were calculated to quantify the discriminant performance of each model. The Delong test was used to determine the significance of the AUC difference among the 12 machine learning models. For the model with the best performance, we further carried out cross-validation tests (3 folds and 5 repeats) to calculate its AUC values in the validation cohort. Additionally, SHapley additive exPlanations (SHAPs) values were introduced to show the importance of each features in the model.


**Statistical Analysis**


The Shapiro–Wilk and Bartlett tests were applied to test the distribution of the clinical variables, then the Student’s t-test, F test, or chi-squared test were applied to determine the between-group differences of the training and validation cohorts, and that of patients with ruptured and unruptured intracranial aneurysms in the training and validation cohorts, respectively. The Spearman correlation analysis was used to evaluate the correlation between optimal radiomics characteristics and the size measurement of aneurysms. A two-tailed *p* < 0.05 was considered statistically significant.

## 3. Results


**Demographics and Clinical Variables**


The demographic information of the patients is shown in Table 1. In training and validation cohorts, patients with ruptured aneurysms were significantly younger, had larger aneurysms, and had higher PHASES scores than patients with unruptured aneurysms. In addition, there were significant differences in the aneurysm location between ruptured and unruptured patients. In the training cohort, the proportion of patients with hypertension was significantly less in the ruptured aneurysm group (*p* = 0.047). Moreover, age, gender, proportion of hypertension, aSAH, aneurysm size, and PHASES scores did not significantly differ between the training and validation cohorts (Appendix A).


**Optimal Radiomics Features**


A total of 107 features of aneurysms were obtained and 21 of them survived the Student’s *t*-test. Secondly, the elastic network regression was used to select the features that were mostly relevant to the rupture of aneurysms; then, seven optimal radiomics features were finally determined (Figure 2). Definitions of the seven optimal radiomics features were described in the Appendix A. Among them, six optimal radiomics features were significantly associated with aneurysm size, with details described in Appendix A Appendix A.


**Machine Learning Model Construction and Evaluation**


The calibration curves of the 12 machine learning models for the probability of ruptured aneurysms showed a good agreement between the actual observed and predicted ruptured aneurysms in the validation cohort (Figure 3A).

The ROC curves of 12 machine learning models in the validation cohort are shown in Figure 3B, and their diagnostic performances are shown in Table 2. With the Delong test, we observed that AdaBoost, gradient boosting, and CatBoost had significantly higher AUCs than the other nine models, among which no significant differences in the AUC were observed (see Appendix A). The radiomics model based on AdaBoost showed the best performance, with an AUC of 0.889 (0.842–0.936) and a sensitivity of 0.716 (0.591–0.817). In addition, the AUCs of the radiomics model based on AdaBoost in the cross-validation tests ranged from 0.842 to 0.913 (Table 3), and the SHAP values of features in the AdaBoost model presented the contribution of seven features to the prediction (Figure 2B), of which the top-three important features were the dependence entropy, elongation, and cluster shape.

## 4. Discussion

In this work, we included patients with ruptured or unruptured intracranial aneurysms, and further constructed prediction models with 12 common machine learning algorithms. We observed that, with the same dataset and radiomics features, all 12 radiomics models based on different algorithms achieved considerable prediction efficacy; however, the efficacy differed across algorithms. 

A previous study reported that the introduction of radiomics features into prediction models could exert a significant improvement on prediction strength with the AUC ranging from 0.767–0.879 (8). Most algorithms we adopted in this work were applied in previous studies individually, including the KNN with feature fusion [21], SVM with aneurysms after surgery with a stent [22], logistic regression with hundreds of features [18], SVM, random forest, logistic regression, and multilayer perceptron with radiomics features and hemodynamics [23], which showed that SVM performed best among them, but without boosting algorithms included. In our study, eight of the 12 models achieved comparable prediction efficiency to those in previous studies with AUC values over 0.80, which indicated that the prediction efficacy of the models we constructed reached the general level. Moreover, our current work included relatively larger bands of machine learning algorithms, and all predictive models were constructed with the same radiomics features and were evaluated on the same basis.

As we established, a larger aneurysm size is related to an increased risk of rupture. However, there still exists a considerable proportion of aneurysms whose diameters larger than 3 mm remained unruptured. This caused us to wonder whether there are more diagnostic indicators that, other than the size of the aneurysm, can be used to predict aneurysm status. Interestingly, six of the seven optimal radiomics features were found to be significantly correlated to the size of the aneurysm. Such radiomics features we employed might have covered the factor of aneurysm size and also information other than regular observations, which might be a crucial aspect of why the application of radiomics would improve the sensitivity of rupture predictions. Additionally, the SHAP analysis indicated that the AdaBoost model tended to correlate heterogeneous density within the aneurysm and the asymmetry of the aneurysm shape with an increased rupture risk. Meanwhile, during the review of cases that suffered false-negative predictions, we observed several cases whose diameters were smaller than 3 mm; however, those with slightly higher values of radiomics features than the average (2 or 3) still suffered from aSAH in the follow up. When observing the original images, we found they were located where the artery bent sharply. After consulting the neurosurgeon, we hypothesized that the rupture might be associated with hemodynamics, and there was a study that included the measurements of hemodynamics, which achieved salient prediction efficacy [23,24]. 

The machine learning algorithms we employed could be classified into three main categories: four in supervised learning (SVM, decision-making tree, logistic regression, and GNB), one in statistical inference (KNN), and the remaining seven in ensemble learning. From the Delong test, we observed that the ensemble learning algorithms were relatively better than those of the others in this study, which might have resulted from the different ways of learning. The algorithms in the supervised learning category were the application of single algorithm classifications, similar to the KNN of the statistical inference, while the algorithms of the ensemble learning integrated better than one algorithm that fit [15]. In a simplified understanding, the random forest could be likened to the effective combination of several decision-making trees. Correspondingly, we might assume that for multifaceted radiomics features displaying different aspects of the aneurysms, ensemble learning could achieve a better performance. It should be noted that a previous study implementing five machine learning algorithms using morphological parameters, including random forest, CatBoost, SVM, light GBM, and XGB, showed a better performance than PHASE, and the SVM was superior among them [25]. Combined with our findings, it might indicate that the performance of the algorithm differs among different types of features. 

In the seven machine learning algorithms of the ensemble learning, AdaBoost showed the highest AUC; though, it was not significantly different from gradient boosting and CatBoost. The best-known application of the AdaBoost algorithm is automatic face recognition, which technically performs multiple iterations on the features of the same training set at different scales [26], and integrates the classifiers generated by each iteration for the final classifier [27,28]. Technically, the boosting algorithm involved a similar technical path, which involves re-iterating the misclassified samples and further optimizes the model [29], which might indicate that machine learning with boosting algorithms would be the better choice for intracranial aneurysm rupture identification based on radiomics. 

This work had several limitations. First, although we used the baseline data from two centers, as well as the follow-up data from ruptured intracranial aneurysms, selection bias was difficult to completely avoid. Second, given the retrospective nature of this study, the timing of the aneurysm rupture was not considered in the analysis, and the diagnostic accuracy may also have been overestimated. Third, although we performed image normalization and a feature homogenization process in the study, there were parameter differences between different CT scanners. Differences beyond causes of biological effects may have also existed. Lastly, given the lack of recognized automatic extraction methods, the method of manually delineating VOIs on CTA images was feasible in this study; however, it may not be suitable for clinical applications. The result of our study could only provide tendentious references and must be further confirmed with larger, related experiments.

## 5. Conclusions

In conclusion, the radiomics models based on the machine learning algorithms were effective computer-aided tools for predicting ruptured intracranial aneurysms, and the AdaBoost algorithm might be superior in the application of radiomics combined with the machine learning algorithm to predict aneurysm ruptures.

## Figures and Tables

**Figure 1 diagnostics-13-02627-f001:**
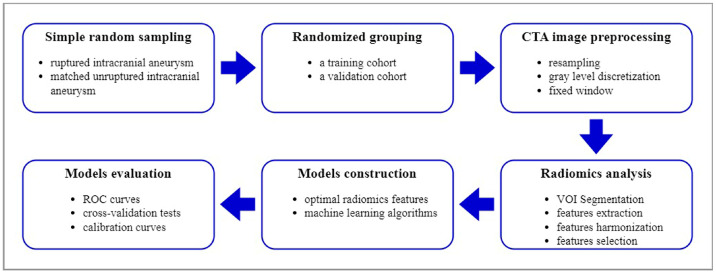
Radiomics combined with machine learning methods to predict the rupture of intracranial aneurysms.

**Figure 2 diagnostics-13-02627-f002:**
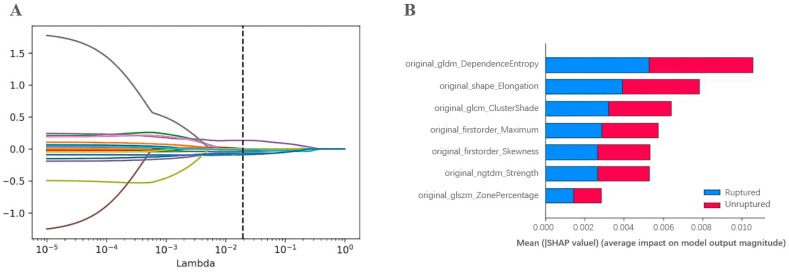
Feature selection with the elastic network regression. (**A**) The elastic network regression coefficient analysis of the 21 radiomics features. Each colored line represents the coefficients of each feature. (**B**) The SHAP values of the 7 optimal radiomics features for the model construction.

**Figure 3 diagnostics-13-02627-f003:**
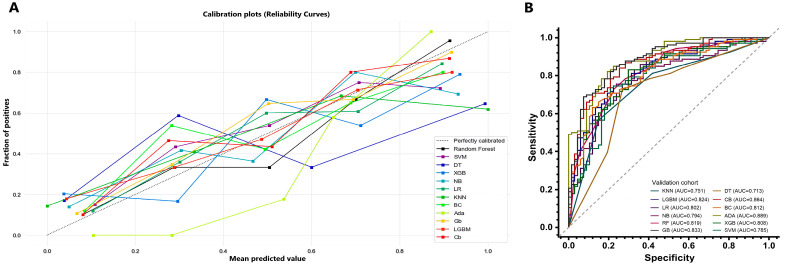
The calibration and receiver operating characteristic curves of models based on 12 machine learning algorithms in the validation cohort. (**A**) The calibration curves of models based on 12 machine learning algorithms in the validation cohort. (**B**) The receiver operating characteristic curves of models based on 12 machine learning algorithms in the validation cohort. Abbreviations: SVM, support machine learning; DT, decision-making tree; XGB, eXtreme gradient boosting; LR, logistic regression; RF, random forest; KNN, k-nearest neighbor; BC, bagging classifier; ADA, AdaBoost; GB, gradient boosting; LGBM, light gradient boosting machine; CB, CatBoost.

**Table 1 diagnostics-13-02627-t001:** Univariates analysis of patients in the training and validation cohorts.

Variables	Training Cohort		*p*-Value	Validation Cohort		*p*-Value
	Ruptured Aneurysm (n = 125)	Unruptured Aneurysm (n = 278)		Ruptured Aneurysm (n = 67)	Unruptured Aneurysm (n = 106)	
Age, y, mean ± SD	58.28 ± 12.04	64.12 ± 11.88	<0.001	58.37 ± 11.76	64.23 ± 13.18	0.003
Gender			0.190			0.533
Male, n (%)	45 (36.0)	120 (43.2)		27 (40.3)	48 (45.3)	
Female, n (%)	80 (64.0)	158 (56.8)		40 (59.7)	58 (54.7)	
Hypertension, n (%)	66 (52.8)	177 (63.7)	0.047	34 (50.7)	69 (65.1)	0.080
aSAH history, n (%)	1 (0.8)	0 (0)	0.310	0 (0)	0 (0)	-
Aneurysm size, mm, median [IQR]	4.8 [3.7, 6.5]	2.6 [2.0, 3.7]	<0.001	4.9 [3.8, 6.8]	2.6 [2.1, 3.5]	<0.001
PHASES score, median [IQR]	4.0 [2.0, 5.0]	1.0 [1.0, 2.0]	<0.001	4.0 [1.0, 5.0]	1.0 [0.8, 2.0]	<0.001
Location			<0.001			<0.001
ICA, n (%)	54 (43.2)	226 (81.3)		29 (43.3)	90 (84.9)	
ACA/ACOM, n (%)	32 (25.6)	16 (5.8)		18 (26.9)	6 (5.7)	
MCA, n (%)	27 (21.6)	21 (7.6)		12 (17.9)	5 (4.7)	
PCOM/Posterior circulation, n (%)	12 (9.6)	15 (5.4)		8 (11.9)	5 (4.7)	

Data are noted as median and interquartile ranges or numbers and percentages in parenthesis. SD, standard deviation; aSAH, aneurysmal subarachnoid hemorrhage; ICA, internal carotid artery; ACA, anterior cerebral artery; MCA, middle cerebral artery; ACOM, anterior communicating artery; PCOM, posterior communicating artery.

**Table 2 diagnostics-13-02627-t002:** The values of prediction indicators of machine learning models in validation cohort.

Validation Cohorts	AUC	Sensitivity	Specificity	PPV	NPV	Predictive Accuracy
RF	0.819 (0.754–0.883)	0.552 (0.426–0.672)	0.877 (0.796–0.930)	0.740 (0.594–0.850)	0.756 (0.669–0.827)	0.751 (0.682–0.810)
SVM	0.785 (0.713–0.856)	0.552 (0.426–0.672)	0.877 (0.796–0.930)	0.740 (0.594–0.849)	0.756 (0.669–0.827)	0.751 (0.682–0.810)
DT	0.713 (0.631–0.795)	0.493 (0.370–0.616)	0.811 (0.721–0.878)	0.623 (0.479–0.749)	0.717 (0.626–0.793)	0.688 (0.615–0.752)
XGB	0.808 (0.742–0.875)	0.627 (0.500–0.739)	0.849 (0.763–0.909)	0.724 (0.589–0.830)	0.783 (0.694–0.852)	0.763 (0.694–0.821)
GNB	0.794 (0.726–0.861)	0.627 (0.500–0.739)	0.811 (0.721–0.878)	0.677 (0.545–0.787)	0.775 (0.684–0.846)	0.740 (0.670–0.800)
LR	0.802 (0.733–0.870)	0.552 (0.426–0.672)	0.887 (0.807–0.938)	0.755 (0.608–0.862)	0.758 (0.671–0.828)	0.757 (0.688–0.815)
KNN	0.751 (0.676–0.826)	0.582 (0.455–0.699)	0.811 (0.721–0.878)	0.661 (0.525–0.776)	0.754 (0.663–0.828)	0.722 (0.651–0.784)
BC	0.812 (0.748–0.877)	0.597 (0.470–0.713)	0.849 (0.763–0.909)	0.714 (0.576–0.823)	0.769 (0.680–0.840)	0.751 (0.682–0.810)
ADA	0.889 (0.842–0.936)	0.716 (0.591–0.817)	0.868 (0.785–0.923)	0.774 (0.647–0.867)	0.829 (0.743–0.891)	0.809 (0.744–0.861)
GB	0.883 (0.833–0.934)	0.657 (0.530–0.766)	0.896 (0.818–0.945)	0.800 (0.720–0.870)	0.805 (0.720–0.870)	0.804 (0.738–0.856)
LGBM	0.824 (0.761–0.886)	0.597 (0.470–0.713)	0.868 (0.785–0.923)	0.741 (0.601–0.846)	0.773 (0.685–0.843)	0.763 (0.694–0.821)
CB	0.864 (0.809–0.919)	0.627 (0.500–0.739)	0.887 (0.807–0.938)	0.778 (0.641–0.875)	0.790 (0.704–0.857)	0.786 (0.719–0.841)

All values are presented with 95% confidence intervals. Abbreviations: AUC, area under curve; PPV, positive predictive value; NPV, negative predictive value. RF, random forest; SVM, support machine learning; DT, decision-making tree; XGB, eXtreme gradient boosting; GNB, Gaussian Naive Bayes; LR, logistic regression; KNN, k-nearest neighbor; BC, bagging classifier; ADA, AdaBoost; GB, gradient boosting; LGBM, light gradient boosting machine; CB, CatBoost.

**Table 3 diagnostics-13-02627-t003:** The results of cross-validation on AdaBoost.

**Folds, Repeats**	Scores	Area Under Curves
1, 1	0.807	0.896
1, 2	0.781	0.863
1, 3	0.766	0.842
1, 4	0.807	0.879
1, 5	0.771	0.862
2, 1	0.833	0.898
2, 2	0.776	0.865
2, 3	0.844	0.904
2, 4	0.865	0.910
2, 5	0.813	0.918
3, 1	0.807	0.888
3, 2	0.833	0.898
3, 3	0.813	0.897
3, 4	0.807	0.873
3, 5	0.833	0.910

## Data Availability

The data and code that support the findings of this study are available on request from the corresponding author. The data are not publicly available due to privacy or ethical restrictions of Second Affiliated Hospital of Chongqing Medical University.

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
