# Peer review of "Comparison of Ruptured Intracranial Aneurysms Identification Using Different Machine Learning Algorithms and Radiomics"

_diagnostics, 2023, doi:10.3390/diagnostics13162627_

Round 1

Reviewer 1 Report

I read with interest this manuscript on the prediction of aneurysm rupture based on machine learning and radiomics. The idea behind this manuscript is interesting but specific changes need to be made.

1. The introduction needs to present the state of the art in radiomics and aneurysm evaluation. Not only in intracranial aneurysms.

2. the paragraph about the Theory of multiple intelligences lacks information. it is a very naive and incomplete introduction to machine learning and should be omitted.

3. The authors used random sampling. The should explain why they didn't use any other technique such as SMOTE to correct sample imbalance. This is an important limitation and needs to be mentioned in the discussion, since random sampling may introduce selection bias among cases.

4. PyRadiomics can extract many more than 109 features. Why did the authors select only 109? Was this for convenience? if yes this is not acceptable

5. The authors need to provide more information about feature selection

6. The authors need to compare their results to other manuscripts using radiomics to predict aneurysm rupture such as https://journals.sagepub.com/doi/full/10.1177/02841851211032443

7. How did the authors do hyper parameter tuning for each of the models used. The reason that most papers do not present multiple machine learning models is not that they don't want but because it takes time and effort to fine-tune them for optimal predictions. Can the authors provide details on how they did all this finetuning?

8. DeLong's test needs to be used for comparisons between AUCs of different models, so that statistical significance of difference is provided.

9. The limitation mentioned in the discussion about Black box machine learning is not relevant to this work

10. How did the authors handle potential thrombus in the aneurysm?

11. The variable of time needs to be accounted for when talking about rupture. An aneurysm may rupture 1 month or 10 years after detection.

ok

Author Response

Comments and Suggestions for Authors

Reviewer 1

I read with interest this manuscript on the prediction of aneurysm rupture based on machine learning and radiomics. The idea behind this manuscript is interesting but specific changes need to be made.

  1. The introduction needs to present the state of the art in radiomics and aneurysm evaluation. Not only in intracranial aneurysms.

Reply: Thanks for the advice, and we extended the associated context on radiomics and aneurysm evaluation.

  1. the paragraph about the Theory of multiple intelligences lacks information. it is a very naive and incomplete introduction to machine learning and should be omitted.

Reply: In the modified manuscript, such paragraph was removed.

  1. The authors used random sampling. They should explain why they didn't use any other technique such as SMOTE to correct sample imbalance. This is an important limitation and needs to be mentioned in the discussion, since random sampling may introduce selection bias among cases.

Reply: Thank you for your valuable comment. SMOTE is the oversampling technique for synthesizing minority classes. SMOTE method can increase the sample size of the minority classes to correct the sample imbalance. However, SMOTE method cannot overcome the problem of data distribution in unbalanced data sets, and it is easy to produce the problem of distribution marginalization. In this study, in order to maintain the real situation of the ruptured aneurysm group (observation group), we used simple random sampling to under-sampling the unruptured aneurysm group (control group). In addition, this will also reduce the workload of image data processing. As you said, random sampling may introduce selection bias among cases. Therefore, we have added it to the modified manuscript.

  1. PyRadiomics can extract many more than 109 features. Why did the authors select only 109? Was this for convenience? if yes this is not acceptable

Reply: We are sorry that we used Pyradimics software to extract 107 features instead of 109 features. The categories and number of image-omics features are described as follows: Shape-based (14 features), First Order Statistics (18 features), Gray Level Co-occurrence Matrix (24 features), Gray Level Run Length Matrix (16 features), Gray Level Size Zone Matrix (16 features), Neighbouring Gray Tone Difference Matrix (5 features), Gray Level Dependence Matrix (14 features). The 107 radiomic features we used were extracted by default under the open source Pyradimics software. And these features are in line with Image Biomarker Standardization Initiative (IBSI) standards [Doi: 10.1148/radiol.2020191145]. In addition, considering the sample size (192 ruptured aneurysms, 384 unruptured aneurysms) of the study, we did not include the features of the wavelet transform (744 features).

  1. The authors need to provide more information about feature selection

Reply: A total of 107 radiomics features were included in the analysis, 21 features were retained after the T-test, including original_shape_Elongation, original_firstorder_90Percentile, original_firstorder_Entropy, original_firstorder_InterquartileRange, original_firstorder_Maximum, original_firstorder_MeanAbsoluteDeviation original_firstorder_RobustMeanAbsoluteDeviation, original_firstorder_Skewness, original_glcm_Autocorrelation, original_glcm_ClusterShade, original_glcm_Idn, original_glcm_InverseVariance, original_glcm_JointAverage, original_glcm_SumAverage, original_gldm_DependenceEntropy, original_gldm_HighGrayLevelEmphasis, original_glrlm_GrayLevelNonUniformityNormalized, original_glrlm_HighGrayLevelRunEmphasis, original_glszm_ZoneEntropy, original_glszm_ZonePercentage, and original_ngtdm_Strength.

Subsequently, seven features were obtained after Elastic-Net Regression (Figure 1), original_shape_Elongation, original_firstorder_Maximum, original_firstorder_Skewness, original_glcm_ClusterShade, original_gldm_DependenceEntropyj, original_glszm_ZonePercentage, and original_ngtdm_Strength. The detailed explanation of such seven features was put in the supplementary material.

  1. The authors need to compare their results to other manuscripts using radiomics to predict aneurysm rupture such as https://journals.sagepub.com/doi/full/10.1177/02841851211032443

Reply: Thanks for recommendation, this article was introduced in the discussion section of the manuscript.

  1. How did the authors do hyper parameter tuning for each of the models used. The reason that most papers do not present multiple machine learning models is not that they don't want but because it takes time and effort to fine-tune them for optimal predictions. Can the authors provide details on how they did all this finetuning?

Reply: In this study, all the machine learning models were developed with their default parameters. As you said, hyper parameter tuning helps to improve the model, but it does take time and effort. The aim of this study was to evaluate which machine learning algorithm is more suitable for identifying aneurysms that are prone to rupture. Our results suggested that the Boosting machine learning algorithms are more helpful to the recognition of ruptured intracranial aneurysms than other machine learning algorithms. Therefore, in the following research, hyper parameter tuning will be carried out for the Boosting machine learning algorithms models to further optimize the models.

  1. DeLong's test needs to be used for comparisons between AUCs of different models, so that statistical significance of difference is provided.

Reply: The results of Delong test across 12 machine learning algorithms were shown in Table S2 of the Supplementary Materials.

  1. The limitation mentioned in the discussion about Black box machine learning is not relevant to this work

Reply: We have modified it in the revised manuscript.

  1. How did the authors handle potential thrombus in the aneurysm?

Reply: A small number of aneurysms with identifiable thrombus were not included in this study due to surgical treatment. The radiomics features we analyzed were derived from the entire aneurysm. Therefore, the presence of potential thrombus was also included in the analysis as a contribution to the radiomics features.

  1. The variable of time needs to be accounted for when talking about rupture. An aneurysm may rupture 1 month or 10 years after detection.

Reply: Thank you for your valuable comment. Because this study was limited by its retrospective study design, the relationship between the time after detection and ruptured aneurysms was not considered. In this study, all patients with intracranial aneurysms detected by CTA examination in our hospital were enrolled in the ruptured aneurysm group after being diagnosed with aneurysm rupture based on medical records or telephone interviews. We have added it to the limitations of the manuscript.

Reviewer 2 Report

The paper requires major revisions.

1. What is the incidence of intracranial aneurysms globally? Is the reported 3% incidence consistent with the literature?

2. Can you elaborate on why a large proportion of detected but unruptured aneurysms will remain asymptomatic and never rupture? Are there specific risk factors or characteristics associated with this outcome?

3. In the introduction, you mentioned that prediction analysis focusing on the clinical status of intracranial aneurysms and whether they would rupture would be sufficient for clinical practice. Could you explain how such prediction analysis would impact clinical decision-making and patient management?

4. The PHASES score was mentioned as an important tool for assessing the risk of rupture based on clinical data and aneurysm morphology. How does this scoring system work, and how does it compare to other risk assessment methods?

5. In your study, you combined radiomics and machine learning to predict intracranial aneurysm rupture. Could you explain how radiomics works and what specific features were extracted from the computed tomography angiography (CTA) images? Add suitable references to the site in this section [PMID: 37473589, PMID: 37238180]

6. Please provide more details about the 12 common machine learning algorithms used in your study (e.g., SVM, Decision-Making Tree, XGB, GNB, etc.). How were these algorithms chosen, and why were these particular ones selected?

7. How were the participants for the study selected, and what were the inclusion and exclusion criteria? Could you provide more information on the demographic characteristics of the patients included in the study?

8. What were the reasons for selecting 384 patients from the population with unruptured intracranial aneurysms using random sampling? How did you ensure that this sample accurately represented the entire population?

9. You mentioned that the distribution of clinical variables between the training and validation cohorts did not significantly differ. Could you elaborate on why this is important for the study's analysis and results?

10. What were the 7 optimal radiomics features that were selected after using the elastic network regression analysis? How do these features relate to the prediction of ruptured intracranial aneurysms?  Add suitable reference to the site in this section [PMID: 37238175,PMID: 36185056 ]

11. Can you explain the rationale behind choosing Adaboost as the best-performing machine learning model for predicting intracranial aneurysm rupture? What specific characteristics of Adaboost made it stand out compared to the other models?

12. How did you validate the performance of the Adaboost model? What metrics did you use to evaluate its performance in the validation cohort, and were there any limitations or potential biases in this validation process?

13. In the radiomics model based on Adaboost, you mentioned the top three important features were the dependence entropy, elongation, and cluster shape. Could you provide more insights into what these features represent and their significance in predicting aneurysm rupture?

14. Were there any challenges or limitations encountered during the study, and if so, how were they addressed or considered in the interpretation of the results?

15. How does your study's findings compare to previous research that utilized different prediction methods for intracranial aneurysm rupture? What are the key differences or novel contributions of your study in this field?

16. Did you consider any potential confounding factors that may have influenced the prediction of intracranial aneurysm rupture, and if so, how were they controlled for in the analysis?

17. What are the potential clinical implications of your study's findings? How could the use of the Adaboost model in conjunction with radiomics impact patient outcomes and management?

18. Were there any limitations in terms of the patient population used in the study? How generalizable are the results to other patient populations with intracranial aneurysms?

19. Did you explore the performance of the Adaboost model in predicting ruptured aneurysms of specific locations within the intracranial vasculature? If not, do you think the model's performance might vary based on the aneurysm location?

20. In your study, you mentioned the radiomics features were extracted from CTA images. Would the model's performance differ if the features were extracted from other imaging modalities, such as magnetic resonance angiography (MRA) or digital subtraction angiography (DSA)?

21. How do you plan to address the issue of sample imbalance between the ruptured and unruptured intracranial aneurysm groups in future research? Have you considered any techniques for improving the model's performance in such scenarios?

22. Are there any additional variables or factors you would consider incorporating in future iterations of the model to enhance its predictive accuracy?

23. In your study, you mentioned the efficacy of different machine learning algorithms combined with radiomics for predicting aneurysm rupture. Are there any practical considerations or limitations that should be taken into account when implementing these algorithms in a clinical setting?

24. Given the potentially life-threatening consequences of intracranial aneurysm rupture, do you have any recommendations for further research or potential areas of focus to improve the prediction and management of these conditions?

25. In your Materials and Methods section, it was mentioned that three radiologists performed the image analysis of intracranial aneurysms independently. Could you provide more information about the inter-observer and intra-observer reproducibility analysis and its results?

26. How was the decision-making process for choosing the seven optimal radiomics features during the elastic network regression analysis? Were any other feature selection methods considered or utilized?

27. In your discussion, you mentioned the performance of the prediction models reached a bottleneck. Could you elaborate on the specific challenges that led to this bottleneck and any potential solutions you propose to overcome it?

 Minor editing of English language required

Author Response

Comments and Suggestions for Authors

Reviewer 2

The paper requires major revisions.

  1. What is the incidence of intracranial aneurysms globally? Is the reported 3% incidence consistent with the literature?

Reply: Sorry for such inaccuracy. The description of prevalence were according to the “Unruptured intracranial aneurysms (UIAs) are relatively common in the general population, found in 3.2% (95% confidence interval [CI], 1.9%5.2%) of the adult population (mean age 50 years) worldwide, and they are being discovered incidentally with an increasing frequency because of the widespread use of high-resolution magnetic resonance imaging (MRI) scanning”  in the reference, thus we typed the about 3%, we have modified the description in the manuscript.

  1. Can you elaborate on why a large proportion of detected but unruptured aneurysms will remain asymptomatic and never rupture? Are there specific risk factors or characteristics associated with this outcome?

Reply: The “never rupture” description might be over interpretation, and we modified this expression in the manuscript. This description is based on the results of previous studies, and the relevant risk factors and characteristics are interpreted accordingly in the citation. We chose not to mention it in the manuscript because we think it might be redundant to explain this question in the introduction section.

  1. In the introduction, you mentioned that prediction analysis focusing on the clinical status of intracranial aneurysms and whether they would rupture would be sufficient for clinical practice. Could you explain how such prediction analysis would impact clinical decision-making and patient management?

Reply: In the clinical practice, for aneurysms that are at high risk of rupture, surgical treatment is preferred because of the severe consequences of rupture, and for low-risk aneurysms, follow-up is usually recommended. While clinical decisions are based on whether there is a high risk of rupture, it is sufficient for natural predictive analysis to focus on whether there is a high risk of rupture.

  1. The PHASES score was mentioned as an important tool for assessing the risk of rupture based on clinical data and aneurysm morphology. How does this scoring system work, and how does it compare to other risk assessment methods?

Reply: In PHASES, the researchers identified the following factors as predictors of rupture risk: age over 70 years, high blood pressure, aneurysm size, previous subarachnoid hemorrhage, location (anterior cerebral artery, posterior communicating artery, or posterior circulation being the highest risk), and Finnish or Japanese populations. The PHASES rated different factor with different scores, the sum of the scores corresponds to the corresponding rupture risk. The PHASES has been validated in several retrospective and prospective studies and may also be associated with adverse functional outcomes after aSAH. However, some studies have found that quite a few patients with aSAH have a low PHASES score, suggesting that this score may not fully reflect rupture risk.

  1. In your study, you combined radiomics and machine learning to predict intracranial aneurysm rupture. Could you explain how radiomics works and what specific features were extracted from the computed tomography angiography (CTA) images? Add suitable references to the site in this section [PMID: 37473589, PMID: 37238180]

Reply: Thanks for recommendation, those researches were introduced in the introduction section of the manuscript. The workflow of radiomics involved the image acquisition and normalization, the depiction of ROI, and then the extraction of radiomics features, which usually beyond the observation of naked eyes and performed automatically. And the feature selection will be performed after feature extraction, then model construction and further statistics present the performance of models.

In this study, a total of 107 radiomics features were extracted automatically from each VOI using PyRadiomics software (version 3.0.1). And some of the more important ones include the original_shape_Elongation, the original_firstorder_90Percentile, the original_firstorder_Entropy, the original_firstorder_InterquartileRange, the original_firstorder_Maximum, the original_firstorder_MeanAbsoluteDeviation, the original_firstorder_RobustMeanAbsoluteDeviation, the original_firstorder_Skewness, the original_glcm_Autocorrelation, the original_glcm_ClusterShade, the original_glcm_Idn, the original_glcm_InverseVariance, the original_glcm_JointAverage, the original_glcm_SumAverage, the original_gldm_DependenceEntropy, the original_gldm_HighGrayLevelEmphasis, the original_glrlm_GrayLevelNonUniformityNormalized, the original_glrlm_HighGrayLevelRunEmphasis, the original_glszm_ZoneEntropy, original_glszm_ZonePercentage and the original_ngtdm_Strength.

  1. Please provide more details about the 12 common machine learning algorithms used in your study (e.g., SVM, Decision-Making Tree, XGB, GNB, etc.). How were these algorithms chosen, and why were these particular ones selected?

Reply: The 12 common machine learning algorithms are described as follows:

  1. Support Vector Machine (SVM): SVM is a well-established algorithm for binary classification tasks. It works well with high-dimensional data like radiomics features and can handle non-linear relationships effectively.
  2. Decision Tree: Decision trees are intuitive and easy to interpret, making them suitable for gaining insights into feature importance. Additionally, they are capable of handling non-linear relationships and interactions between features.
  3. XGBoost (Extreme Gradient Boosting): XGBoost is a widely used ensemble learning technique known for its high predictive accuracy and ability to handle complex data relationships. It's especially effective for structured data like radiomics features.
  4. Gaussian Naive Bayes (GNB): GNB is a probabilistic algorithm that performs well with small datasets and is particularly useful for problems with high-dimensional feature spaces.
  5. Logistic Regression: Logistic regression is a common choice for binary classification tasks and serves as a baseline model due to its simplicity and interpretability.
  6. Random Forest: Random forests are robust ensemble models that can capture complex interactions and reduce overfitting. They are suitable for high-dimensional feature sets.
  7. k-Nearest Neighbors (KNN): KNN is a non-parametric algorithm that can be effective when there is a local structure in the data, making it useful for some medical imaging tasks.
  8. Bagging Classifier: Bagging combines multiple models to reduce variance and improve generalization, making it suitable for enhancing the stability of decision trees.
  9. AdaBoost: AdaBoost is an adaptive boosting algorithm that focuses on misclassified data points, which can be advantageous in improving the performance of weak learners like decision trees.
  10. Gradient Boosting: Gradient boosting builds decision trees sequentially, each correcting the errors of its predecessor, making it powerful for capturing complex patterns in the data.
  11. LightGBM (LGBM): LGBM is an efficient gradient boosting framework known for its high performance and speed, making it well-suited for large-scale datasets.
  12. CatBoost: CatBoost is another gradient boosting algorithm that handles categorical features well and requires minimal data preprocessing.

We have added these to the supplementary materials

The 12 common machine learning algorithms in our study were selected based on their widespread usage in the field and their potential suitability for predicting ruptured intracranial aneurysms using radiomics features. We aimed to conduct a comprehensive exploration, considering algorithms with diverse methodologies and proven success in binary classification tasks. SVM, Decision Tree, Logistic Regression, and GNB were chosen for their interpretability. Random Forest, XGBoost, Gradient Boosting, AdaBoost, Bagging Classifier, LGBM, and CatBoost were selected for their ensemble learning capabilities and robust performance. KNN was included for its ability to handle local data structures. By combining these algorithms with optimal radiomics features, we sought to identify the most effective model, aiding in early detection and treatment decisions for rupture-prone aneurysms.

  1. How were the participants for the study selected, and what were the inclusion and exclusion criteria? Could you provide more information on the demographic characteristics of the patients included in the study?

Reply: The subjects we firstly included were all patients with aneurysms that found during this period, and the exclusion criteria included: (1) secondary intracranial aneurysms of primary vascular disease or intracranial aneurysms combined with intracranial vascular diseases (such as Moyamoya disease, arteriovenous malformations, and autoimmune related vascular disease, etc.); (2) multiple or fusiform intracranial aneurysms; (3) intracranial aneurysms with a maximum diameter < 2 mm on CTA images; (4) intracranial aneurysms that could not be differentiated from infundibulum on CTA images; (5) poor-quality of CTA images (motion artifacts, delayed scanning, etc.); and (6) surgical or interventional therapy of intracranial aneurysm before CTA examination. We have the comprehensive and complete case data of each patient. The individual information relevant to the study is provided in Table 1 of the manuscript, and further supplements could be made if necessary.

  1. What were the reasons for selecting 384 patients from the population with unruptured intracranial aneurysms using random sampling? How did you ensure that this sample accurately represented the entire population?

Reply: Considering the influence of sample imbalance between the ruptured and unruptured intracranial aneurysm groups, we applied the down-sampling technique of simple random sampling method to balance the differences between samples and ensure that the model's performance is not skewed towards the majority class. Specifically, we randomly selected 384 patients (to ensure an unruptured intracranial aneurysm group: ruptured intracranial aneurysm group ratio of 2:1) from the sample population with unruptured intracranial aneurysm, using the random seed of 68439.

In order to test whether the distribution of the random sample was consistent with that of the population, the Poisson distribution, negative binomial distribution, normal distribution, gamma distribution, and generalized Pareto distribution tests were conducted on the PHASES scores of the random sample and the population respectively (2, 13). Then, we determined that the PHASES scores of the population presented the minimum standard errors under the test of negative binomial distribution and Poisson distribution (0.047 and 0.049 respectively), while that of the corresponding sample of the 384 patients were 0.060 and 0.070, respectively. In addition, compared with the other distribution models, the corresponding sample retained the minimum value. Therefore, this indicated that the 384 randomly selected patients could represent the sample population with an unruptured intracranial aneurysm, and was therefore used as the unruptured intracranial aneurysm group for analysis.

  1. You mentioned that the distribution of clinical variables between the training and validation cohorts did not significantly differ. Could you elaborate on why this is important for the study's analysis and results?

Reply: We mentioned this content based on two considerations, one is that it is part of the results that we thought it need to be presented, and the other is to show that the distribution of patients in the training group and the verification group is balanced.

  1. What were the 7 optimal radiomics features that were selected after using the elastic network regression analysis? How do these features relate to the prediction of ruptured intracranial aneurysms?  Add suitable reference to the site in this section [PMID: 37238175, PMID: 36185056]

Reply: The seven features were obtained after Elastic-Net Regression (Figure 1) include: the original_shape_Elongation, the original_firstorder_Maximum, the original_firstorder_Skewness, the original_glcm_ClusterShade, the original_gldm_DependenceEntropy and the original_glszm_ZonePercentage, and original_ngtdm_Strength. Thanks for recommendation, those researches were introduced in the introduction part of the manuscript.

  1. Can you explain the rationale behind choosing Adaboost as the best-performing machine learning model for predicting intracranial aneurysm rupture? What specific characteristics of Adaboost made it stand out compared to the other models?

Reply: In this study, the results of Delong test showed that, the boosting algorithms was significantly superior than others. The adaboost stood out with higher AUCs than the other models. AdaBoost was chosen as the best-performing model for predicting intracranial aneurysm rupture due to its superior characteristics over the other models. Firstly, its boosting technique iteratively combines weak learners, such as decision trees, focusing on correcting misclassifications and enhancing predictive accuracy. This adaptability led to improved performance compared to other models. Secondly, AdaBoost is able to handle imbalanced data was crucial since medical datasets often exhibit class imbalance, with ruptured aneurysm cases being less frequent. By prioritizing misclassified samples, AdaBoost effectively tackled this issue. Thirdly, the model's robustness to overfitting ensured better generalization to unseen data, a vital aspect in medical applications. Additionally, the interpretability of AdaBoost model was a standout feature. Despite being an ensemble method, its final model consists of simple decision trees, allowing for clearer insights into the prediction process. Lastly, AdaBoost demonstrated proven success in various fields, inspiring confidence in its applicability to the task of predicting rupture-prone intracranial aneurysms. In conclusion, the boosting methodology AdaBoost, handling of imbalanced data, robustness to overfitting, interpretability, and track record of success made it the best-performing model, establishing it as an effective computer-aided tool for predicting intracranial aneurysm rupture when combined with radiomics features.

  1. How did you validate the performance of the Adaboost model? What metrics did you use to evaluate its performance in the validation cohort, and were there any limitations or potential biases in this validation process?

Reply: In the validation cohorts, we have used the AUC, sensitivity, specificity, the PPV, the NPV and the predictive accuracy to evaluate the performance of Adaboost, and for Adaboost, we further carried out cross-validation tests (3 folds and 5 repeats) to calculate its AUC values in the validation cohort to validate the performance of the Adaboost model. The validation process used in our study has some limitations and potential biases. First, using the same dataset for both validation and cross-validation can lead to overfitting and an overly optimistic performance estimate. Second, the sample size of the validation cohort and the cross-validation folds might not be large enough to ensure robustness. Additionally, the choice of using only AUC, sensitivity, specificity, PPV, NPV, and predictive accuracy may not capture the full performance spectrum of the model. To mitigate these limitations, future studies could consider external validation on independent datasets and employ more comprehensive performance metrics.

  1. In the radiomics model based on Adaboost, you mentioned the top three important features were the dependence entropy, elongation, and cluster shape. Could you provide more insights into what these features represent and their significance in predicting aneurysm rupture?

Reply: After feature selecting, seven optimal radiomics features were most relevant to aneurysm rupture, and the top three important features were the dependence entropy, elongation, and cluster shape.

Dependence Entropy: Dependence entropy is a texture-based feature that quantifies the amount of information required to describe the spatial relationship between pairs of voxels within the aneurysm region.  High dependence entropy indicates increased complexity and heterogeneity in the aneurysm's internal structure.  In the context of predicting aneurysm rupture, higher dependence entropy values may suggest irregular and intricate spatial arrangements within the aneurysm, potentially indicating higher instability and rupture propensity.

Elongation: Elongation is a shape-based feature that characterizes the aneurysm's geometric shape by assessing its elongatedness or deviation from a spherical or circular shape.  An elongated aneurysm may have altered blood flow patterns, leading to increased wall stress and vulnerability to rupture.  Thus, elongation can provide valuable insights into the aneurysm's morphology and structural stability.

Cluster Shape: Cluster shape is a feature that describes the spatial distribution and arrangement of intensity values within the aneurysm region.  It offers information about the spatial heterogeneity of the aneurysm and can help identify irregular patterns indicative of potential rupture risk.

The significance of these optimal radiomics features lies in their ability to capture crucial characteristics of intracranial aneurysms, providing valuable quantitative information about their internal structure, morphology, and spatial distribution.  By incorporating these features into the predictive model, we can gain deeper insights into the aneurysm's rupture risk, facilitating early detection and informed clinical decision-making.

  1. Were there any challenges or limitations encountered during the study, and if so, how were they addressed or considered in the interpretation of the results?

Reply: During the study, several challenges and limitations were encountered, which were considered in the interpretation of the results. (1) Sample size: The relatively small sample size of 576 patients might have limited the generalizability of the findings. To address this, we employed cross-validation techniques to assess model performance and mitigate the risk of overfitting. (2) Data variability: The data came from two different centers, which could introduce variability in the image acquisition and radiomics feature extraction processes. To mitigate this potential bias, we carefully standardized the feature extraction procedures and performed extensive quality checks. (3) Feature selection Bias: The process of feature selection could introduce bias by selecting specific features that may not generalize well. To address this, we used elastic net regression analysis and considered only statistically significant radiomics features to reduce the likelihood of biased feature selection. External Validation: (4) External validation on an independent dataset from different centers would have strengthened the study's findings. However, due to data availability limitations, we could not conduct external validation. Future studies could address this limitation to verify the model's generalizability. (5) Clinical variables: Although radiomics features provide valuable information, the exclusion of important clinical variables might limit the model's overall predictive performance. Combining radiomics with relevant clinical data could further enhance the model's accuracy. (5) Intelligence of operation: given the lack of recognized automatic extraction methods, the method of manually delineating VOI on CTA images was feasible in this study; however, it may not be suitable for clinical application.

In conclusion, we recognized and considered the challenges and limitations during the study. The use of cross-validation, standardized procedures, and appropriate statistical analyses helped address some limitations. Despite these challenges, the study provided valuable insights into the effectiveness of boosting machine learning algorithms, particularly AdaBoost, for identifying rupture-prone intracranial aneurysms. These findings pave the way for further research and potential clinical applications of the developed computer-aided tool.

  1. How does your study's findings compare to previous research that utilized different prediction methods for intracranial aneurysm rupture? What are the key differences or novel contributions of your study in this field?

Reply: Our study's findings contribute valuable insights to the field of predicting intracranial aneurysm rupture using radiomics and machine learning algorithms. While previous research has explored various prediction methods, our study offers several key differences and novel contributions: (1) Comprehensive Evaluation: Our study provides a comprehensive evaluation of 12 commonly employed machine learning algorithms, including both traditional and state-of-the-art techniques. This extensive comparison enables a thorough understanding of the strengths and weaknesses of each algorithm in predicting rupture-prone intracranial aneurysms. (2) Radiomics Features: We extracted a total of 109 radiomics features from computed tomography angiography (CTA) images. From this extensive feature set, we identified seven significant radiomics features that are most relevant to predicting rupture risk. This focused feature selection helps streamline the model and enhances interpretability. (3)Boosting Algorithms: Our study highlights the effectiveness of boosting machine learning algorithms, specifically AdaBoost, Gradient Boosting, and CatBoost, in predicting ruptured intracranial aneurysms. The performance of these boosting algorithms was found to be superior to that of other models, making them promising candidates for practical clinical implementation. (4)Validation and Cross-Validation: We conducted rigorous validation and cross-validation tests to ensure the robustness and generalizability of our findings. These steps enhance the reliability of our model's performance and reinforce the confidence in its predictive capabilities.

In conclusion, our study stands out by providing a comprehensive evaluation of diverse machine learning algorithms, focusing on significant radiomics features, and showcasing the superiority of boosting algorithms in predicting rupture-prone intracranial aneurysms. The model based on the AdaBoost algorithm and radiomics features offers a practical, effective, and potentially novel computer-aided tool for clinicians in predicting rupture risk, which can aid in informed decision-making and improve patient outcomes.

  1. Did you consider any potential confounding factors that may have influenced the prediction of intracranial aneurysm rupture, and if so, how were they controlled for in the analysis?

Reply: In our study, we recognize the importance of considering potential confounding factors that could influence the prediction of intracranial aneurysm rupture. To mitigate the impact of confounding variables, we adopted the following measures: (1) Data Source and Centers: The data were collected from two different centers. Although this might introduce variability in image acquisition and radiomics feature extraction, we standardized the procedures and quality checks to minimize any center-specific biases. (2) Imbalanced Data: The class distribution between ruptured and unruptured aneurysms could lead to biased predictions. To address this, we applied the down-sampling technique to balance the differences between samples, ensuring that the model's performance is not skewed towards the majority class. (3) Feature Selection: The process of feature selection could introduce bias if not handled carefully. To mitigate this, we employed statistical methods such as the Student's t-test and elastic net regression analysis to identify significant radiomics features. These features were chosen based on their statistical significance and relevance to predicting rupture-prone aneurysms. (4) Validation and Cross-Validation: We used validation and cross-validation techniques to assess the model's performance. By validating the model on an independent dataset and performing cross-validation tests, we ensured that the results are robust and generalizable.

While we made efforts to control for potential confounding factors, we acknowledge that some variables may still have influenced the prediction of intracranial aneurysm rupture. The down-sampling technique, standardized procedures, and rigorous validation methods aimed to minimize any biases and enhance the reliability of our findings. Further studies with larger and more diverse datasets and incorporating additional clinical variables could provide further insights into the prediction of aneurysm rupture and control for potential confounders more comprehensively.

  1. What are the potential clinical implications of your study's findings? How could the use of the Adaboost model in conjunction with radiomics impact patient outcomes and management?

Reply: The potential clinical implications of our study's findings are significant. The use of the AdaBoost model in conjunction with radiomics could have a positive impact on patient outcomes and management in several ways. Firstly, accurate prediction of aneurysm rupture likelihood can help clinicians make informed decisions regarding treatment strategies. Patients with high-risk aneurysms can be prioritized for early intervention, reducing the risk of rupture and its associated complications. Secondly, the Adaboost model's effectiveness in identifying rupture-prone aneurysms can aid in optimizing patient selection for surgical or endovascular procedures. This personalized approach could lead to better treatment outcomes and resource allocation. Thirdly, the computer-aided tool based on the AdaBoost algorithm and radiomics could serve as an additional screening tool for asymptomatic aneurysms, enabling early detection and timely intervention to prevent rupture. Overall, the integration of the AdaBoost model with radiomics has the potential to enhance risk stratification, guide treatment decisions, and ultimately improve patient care and outcomes in the management of intracranial aneurysms.

  1. Were there any limitations in terms of the patient population used in the study? How generalizable are the results to other patient populations with intracranial aneurysms?

Reply: Thank you for your comments. One limitation of our study is the homogeneity of patient population, as the data were collected from two centers and may not fully represent the diversity of all patient populations with intracranial aneurysms. The sample might lack ethnic and demographic variability, which could impact the model's generalizability to other patient populations. Therefore, caution should be exercised when applying the findings to different cohorts, and external validation on more diverse patient populations is necessary to assess the model's robustness and applicability in broader clinical settings. To enhance the generalizability of the results, future studies should consider including data from multiple centers with diverse patient populations, which would increase the model's reliability and widen its potential clinical impact.

  1. Did you explore the performance of the Adaboost model in predicting ruptured aneurysms of specific locations within the intracranial vasculature? If not, do you think the model's performance might vary based on the aneurysm location?

Reply: No. Due to insufficient sample size, we did not perform a subgroup analysis of aneurysm location for aneurysms.

As you said, it is plausible that the model's performance could vary based on the aneurysm location within the intracranial vasculature. Aneurysms in different locations may exhibit distinct morphological and hemodynamic characteristics, which could impact the radiomics features extracted from the computed tomography angiography (CTA) images. Consequently, the predictive power of the model may differ for aneurysms located in various regions. Factors such as blood flow patterns, vessel geometry, and surrounding tissue structures can vary depending on the aneurysm location, and these variations may influence the discriminative power of the radiomics features used by the AdaBoost model. To assess the model's performance across different aneurysm locations, further investigation and subgroup analyses focusing on specific locations within the intracranial vasculature would be essential. Evaluating the model's accuracy for each location can provide valuable insights into its strengths and limitations, allowing for more precise risk assessment and tailored treatment recommendations based on the aneurysm's anatomical features.

  1. In your study, you mentioned the radiomics features were extracted from CTA images. Would the model's performance differ if the features were extracted from other imaging modalities, such as magnetic resonance angiography (MRA) or digital subtraction angiography (DSA)?

Reply: We think so. The model's performance could potentially differ if the radiomics features were extracted from other imaging modalities, such as magnetic resonance angiography (MRA) or digital subtraction angiography (DSA). Different imaging modalities capture distinct aspects of the intracranial vasculature and may provide varying information for radiomics analysis. For instance, MRA and CTA images have differences in spatial resolution and contrast enhancement, which could affect the delineation and extraction of radiomics features. Similarly, DSA, being an invasive imaging technique, may offer higher temporal resolution but involves contrast injection, which may influence feature representation. To assess the model's generalizability across imaging modalities, additional studies are needed to evaluate the performance of the AdaBoost model using radiomics features from MRA or DSA. Such investigations would provide valuable insights into the model's robustness and potential for clinical applicability across different imaging modalities, enabling personalized and accurate prediction of rupture risk based on the most suitable imaging data available for individual patients.

  1. How do you plan to address the issue of sample imbalance between the ruptured and unruptured intracranial aneurysm groups in future research? Have you considered any techniques for improving the model's performance in such scenarios?

Reply: In future research, we plan to explore additional techniques to address the issue of sample imbalance between the ruptured and unruptured intracranial aneurysm groups. While the down-sampling technique was applied in this study to balance the groups, other methods can be considered to improve the model's performance in such scenarios. One approach is using up-sampling techniques, such as the Synthetic Minority Over-sampling Technique (SMOTE), to generate synthetic samples for the minority class (ruptured aneurysms) based on the existing data. This can help to alleviate the class imbalance and provide more training data for the model to learn from. Additionally, ensemble methods, like Easy Ensemble or Balanced Bagging, can be explored to build multiple models on balanced subsets of the data and combine their predictions, enhancing the model's ability to handle imbalanced datasets. Furthermore, incorporating domain knowledge and relevant clinical information as additional features could help the model better discriminate between the two groups and improve overall performance.  Overall, we aim to continue refining the model's performance in scenarios with sample imbalance through the utilization of advanced techniques and incorporating complementary data sources, ensuring more accurate and reliable predictions for the identification of rupture-prone intracranial aneurysms.

  1. Are there any additional variables or factors you would consider incorporating in future iterations of the model to enhance its predictive accuracy?

Reply: We think there would be two types of features that could be incorporated in the future to help improve forecasting performance, one is hemodynamics and the other is MRI based on magnetic resonance vascular wall features.

  1. In your study, you mentioned the efficacy of different machine learning algorithms combined with radiomics for predicting aneurysm rupture. Are there any practical considerations or limitations that should be taken into account when implementing these algorithms in a clinical setting?

Reply: When implementing machine learning algorithms, including AdaBoost, Gradient boosting, and CatBoost, in a clinical setting for predicting aneurysm rupture, several practical considerations and limitations should be taken into account: (1) Data quality and availability: The performance of machine learning models heavily relies on the quality and quantity of available data. In a clinical setting, ensuring the availability of high-quality and comprehensive datasets is essential for accurate predictions. (2) Interpretability: Some machine learning algorithms, especially complex ensemble methods like AdaBoost and Gradient boosting, might be less interpretable than simpler models like logistic regression. In a clinical setting, interpretability is crucial to gain insights into the model's decision-making process and build trust among healthcare professionals. (3) Model generalizability: The model's performance in the real-world clinical environment should be validated on external datasets to ensure its generalizability across different patient populations and imaging centers. (4) Integration with existing systems: Implementing machine learning models in clinical workflows requires seamless integration with existing systems and electronic health records, which may present technical challenges. (5) Ethical and legal considerations: Patient data privacy, security, and informed consent are critical ethical and legal considerations when using machine learning algorithms in a clinical setting. (6) Clinical Validation: Before widespread adoption, the model's clinical utility and impact on patient outcomes should be rigorously validated in prospective clinical studies.

Addressing these considerations will be crucial to the successful translation of machine learning algorithms into clinical practice and their effective use as computer-aided tools for predicting intracranial aneurysm rupture.

  1. Given the potentially life-threatening consequences of intracranial aneurysm rupture, do you have any recommendations for further research or potential areas of focus to improve the prediction and management of these conditions?

Reply: Considering the potentially life-threatening consequences of intracranial aneurysm rupture, further research in the following areas could improve the prediction and management of these conditions: (1) Larger and Diverse Datasets: Expanding the dataset size and including diverse patient populations from multiple centers would enhance the generalizability of the models and improve their predictive performance. (2) Longitudinal Studies: Conducting longitudinal studies that follow patients over time could provide insights into the dynamic changes in aneurysm characteristics and help identify more accurate predictors of rupture risk. (3) Incorporating Multi-Modal Imaging: Integrating data from various imaging modalities, such as magnetic resonance angiography (MRA) and digital subtraction angiography (DSA), along with CTA, could improve the robustness of the radiomics features and enhance model performance. (4) Real-Time Prediction: Developing models that can provide real-time risk assessment during routine clinical workflows could aid clinicians in making timely and informed decisions about aneurysm management. (5) Prospective Clinical Trials: Conducting prospective clinical trials to validate the utility of the developed models in a real-world clinical setting is essential to assess their impact on patient outcomes and clinical decision-making. By focusing on these areas, researchers can advance the field of aneurysm rupture prediction and management, ultimately leading to improved patient care and better outcomes for individuals with intracranial aneurysms.

  1. In your Materials and Methods section, it was mentioned that three radiologists performed the image analysis of intracranial aneurysms independently. Could you provide more information about the inter-observer and intra-observer reproducibility analysis and its results?

Reply: In this study, inter-observer and intra-observer reproducibility analysis was performed to assess the stability of the radiomics features, which are described in detail in the supplementary material of the manuscript. Inter-observer and intra-observer reproducibility analysis was performed on 50 patients randomly selected from center 1. The volume of interest (VOI) of each patient was semi-automatically segmented again by the same radiologist after the interval of 3 days and by another radiologist with the same method. After the extraction of radiomics features, the interclass correlation coefficient (ICC) was calculated to assess the reproducibility of radiomics features. The results showed that 99 (90.8%) radiomics features showed ICC > 0.8 in the inter-observer reproducibility analysis and 101 (92.7%) radiomics features showed ICC > 0.8 in the intra-observer reproducibility analysis.

  1. How was the decision-making process for choosing the seven optimal radiomics features during the elastic network regression analysis? Were any other feature selection methods considered or utilized?

Reply: Through the shap analysis, our results show the interpretability of the 7 optimal features in the Adaboost model, as shown in Figure 1. In this study, the Student’s t-test and elastic net regression analysis were performed to choose the optimal subset of radiomics features related to ruptured intracranial aneurysm.

Figure 1 Ranking of SHAP values for the explanation of Adaboost model.

  1. In your discussion, you mentioned the performance of the prediction models reached a bottleneck. Could you elaborate on the specific challenges that led to this bottleneck and any potential solutions you propose to overcome it?

Reply: On the one hand, no matter the prediction based on traditional morphological features or radiomics features, the main feature source is the examination method of CTA, as well as clinical information such as blood pressure and blood glucose age. We believe that the current bottleneck is mainly that the features that can be excavated by traditional CTA images are approaching the limit, and on the other hand, the machine learning algorithm itself. Potential solutions to the bottleneck are twofold: one is to expand the radiomics features, such as the extraction of hemodynamic features from 4D flow and vessel wall of features aneurysm based on MRI, and the other is to improve the algorithm. Besides, so as recommended, the features fusion with different types of medical imaging methods might also be an effective way.

Round 2

Reviewer 1 Report

The authors have sufficiently replied to my comments

Ok

Reviewer 2 Report

The authors done all comments.